# Short Chain Fatty Acids and Fecal Microbiota Abundance in Humans with Obesity: A Systematic Review and Meta-Analysis

**DOI:** 10.3390/nu11102512

**Published:** 2019-10-18

**Authors:** Kyu Nam Kim, Yao Yao, Sang Yhun Ju

**Affiliations:** 1Department of Family Practice and Community Health, Ajou University School of Medicine, Suwon, Gyeonggi-do 16499, Korea; ktwonm@hanmail.net; 2Center for Healthy Aging and Development Studies and Raissun Institute for Advanced Studies, National School of Development, Peking University, Beijing 100871, China; yaoy301@126.com; 3Department of Family Medicine, Uijeongbu St. Mary’s Hospital, College of Medicine, The Catholic University of Korea, 271, Cheonbo-Ro 11765, Uijeongbu-si, Gyeonggi-do, 11765, Korea

**Keywords:** obesity, microbiome, feces, colon, humans, systematic review

## Abstract

There have been mixed results regarding the relationship among short chain fatty acids (SCFAs), microbiota, and obesity in human studies. We selected studies that provided data on SCFA levels or fecal microbiota abundance in obese and nonobese individuals and then combined the published estimates using a random-effects meta-analysis. Obese individuals had significantly higher fecal concentrations of acetate (SMD (standardized mean differences) = 0.87, 95% CI (confidence interva) = 0.24–1.50, *I*^2^ (*I*–squared) = 88.5), propionate (SMD = 0.86, 95% CI = 0.35–1.36, *I*^2^ = 82.3%), and butyrate (SMD = 0.78, 95% CI = 0.29–1.27, *I*^2^ = 81.7%) than nonobese controls. The subgroup analyses showed no evidence of heterogeneity among obese individuals with a BMI >30 kg/m^2^ (*I^2^* = 0.0%). At the phylum level, the abundance of fecal microbiota was reduced in obese compared to nonobese individuals, but the difference was not statistically significant (Bacteroidetes phylum, SMD = −0.36, 95% CI = −0.73–0.01; Firmicutes phylum, SMD = −0.10, 95% CI = −0.31–0.10). The currently available human case-control studies show that obesity is associated with high levels of SCFA but not gut microbiota richness at the phylum level. Additional well-designed studies with a considerable sample size are needed to clarify whether this association is causal, but it is also necessary to identify additional contributors to SCFA production, absorption, and excretion in humans.

## 1. Introduction

Obesity is a state in which excess fat accumulates in the body due to an imbalance between energy intake and energy consumption [1]. As the prevalence of obesity in adults, adolescents, and children increases globally and the prevalence of obesity-related diseases and mortality rates increases, the numbers of epidemiological and experimental studies are increasing to identify host and environmental factors that affect energy balance [1,2,3]. 

Short chain fatty acids (SCFAs) are organic fatty acids with one to six carbon atoms and are the major anions produced by microbial fermentation of undigested carbohydrates, but the amount of SCFAs depends on various host, environmental, dietary, and gut microbiota factors [2]. Butyrate, propionate, and acetate, which account for 90% to 95% of the SCFAs present in the colon, act as signaling ligands between the gut microbiome and host metabolism at specific levels [2,3,4]. 

Emerging evidence based on numerous animal studies has shown that the gut microbiota and its metabolites, particularly SCFAs, play an important role in obesity [5,6,7]. The results of these studies have indicated that micronutrients and SCFAs produced by intestinal bacteria can regulate host energy metabolism in the development of diet-induced obesity, thereby increasing de novo lipogenesis in the liver and lipid accumulation in all fat stores. However, in human studies, there have been mixed results regarding the relationship between SCFAs and obesity. For example, some studies have reported a positive correlation between fecal SCFA concentrations and obesity [8,9,10]; however, others have reported a negative relationship between SCFA levels and obesity [11]. 

Meanwhile, several studies have shown that gut microbiota compositions associated with the production of SCFAs are also associated with obesity. However, this relationship has not yet been clarified in humans. For example, Furet JP et al. reported an increase in phylum Firmicutes and a decrease in Bacteroidetes associated with obesity [12]; however, Schwiertz et al. reported the opposite association [13] and Duncan SH et al. failed to find the same association [14]. Moreover, a meta-analysis of human studies did not distinguish obese from lean fecal microbiota, which could bring potential heterogeneity [15]. 

Therefore, the main objective of the current meta-analysis based on previous human studies was to investigate whether SCFA levels in obese individuals differ from those in nonobese individuals. The secondary objective of this study was to determine whether the fecal microbiota abundance in obese and nonobese individuals reported in the literature on the relationship between SCFAs and obesity is different.

## 2. Materials and Methods 

We planned, conducted, and reported this systematic review according to widely accepted quality standards for reporting meta-analyses of observational studies in epidemiology and preferred reporting items for systematic reviews and meta-analyses guidelines [16,17]. 

### 2.1. Literature Search

A medical librarian with experience in systematic reviews participated in designing the search strategy. We searched the PubMed, Cochrane Library, and EMBASE databases via Elsevier for reports published between March 1953 and May 2018, and an updated search was performed in May 2019. A PubMed search for studies on SCFAs and obesity was conducted without restrictions by combining search terms that were synonymous with or related to SCFAs and obesity. The keywords used in the PubMed search were converted into search tags for the Cochrane Library and EMBASE databases (Appendix A). Furthermore, the reference lists of relevant articles were manually searched to identify additional studies. We followed all of the recommended standards listed in the meta-analysis of observational studies in epidemiology checklist [17].

### 2.2. Inclusion and Exclusion Criteria

Published articles were included in this meta-analysis if they met the following criteria: (1) Case-control studies that were conducted in humans rather than animals; (2) studies that provided data on SCFA levels in individuals both with and without obesity; and (3) studies that were written in English and published in their entirety. The exclusion criteria in this meta-analysis were the following: (1) Articles that did not satisfy the inclusion criteria; (2) publication types, such as animal studies, reviews, case reports, and systematic reviews; and (3) studies that did not provide sufficient data on SCFA levels, including means, medians, standard deviations (SDs), and/or standard errors for individuals with and without obesity.

### 2.3. Data Extraction and Quality Assessment

Two investigators (Sang Yhun Ju and Kyu Nam Kim), the coauthors of the present study, independently extracted the data from the original reports. The following information was extracted: First author’s last name, year of publication, country, age, sex, sample size, SCFA levels, SCFA assessment methods, fecal microbiota abundance, and definitions of obesity used. Disagreements between the two reviewers were resolved by consensus. The methodological quality of the included studies was evaluated using the Newcastle–Ottawa scale (NOS) criteria for case-control studies, which contained nine items that were grouped into three major categories. The maximum scores were 4 for selection, 2 for comparability, and 3 for exposure. A final score of 7 or more was indicative of a high-quality study.

### 2.4. Assessment of Quality of Evidence

The quality of evidence was evaluated by means of the Grading of Recommendations Assessment, Development, and Evaluation (GRADE) criteria [18]. Two investigators (Sang Yhun Ju and Kyu Nam Kim) independently assessed risk of bias, inconsistency, indirectness, imprecision, and publication bias. Overall quality was graded using the GRADEPro Guideline Development Tool [19].

### 2.5. Data Synthesis and Statistical Analyses

The data of interest, presented as continuous values (means and SDs), were used to perform the meta-analysis to obtain the standardized mean differences (SMDs) and 95% confidence intervals (CIs) of the SCFA levels of participants with obesity and controls. The SMDs were calculated by subtracting the means of SCFAs levels between the two groups and dividing by the standard deviations. Thus, studies for which the difference in means is the same proportion of the standard deviation will have the same SMD, regardless of the actual scales used to make the SCFAs measurements. An SMD below 0.5 was considered small, 0.5–0.8 was considered moderate, and over 0.8 was considered large [20]. Our protocol proposed the pooling of SMDs for the meta-analysis using a random-effects model [21]. The statistical heterogeneity among the studies was assessed using *I^2^* statistics [22]. *I^2^* values greater than 50% indicated high heterogeneity. Heterogeneity was also assessed by comparing the results from studies grouped according to mean age using meta-regression. To evaluate the potential sources of heterogeneity in the analyses, we also conducted subgroup and sensitivity analyses. Publication bias was evaluated visually using Begg’s funnel plot and Egger’s test [23]. In the presence of publication bias, the *p*-values for Egger’s test were less than 0.1. All statistical analyses were performed using Stata software, version 15.0 (Stata Corp., College Station, TX, USA).

## 3. Results

### 3.1. Study Search and Selection and Characteristics of Eligible Studies

Figure 1 shows the details of the study selection process. Briefly, we identified 29 potentially relevant articles on SCFAs in relation to obesity. The interrater reliability of the two reviewers for the initial screening of the study selection was moderate (agreement 86.75%, *κ* = 0.19). After we further examined the 29 identified articles, 22 articles were excluded (Appendix A). Finally, we identified seven articles that met the inclusion criteria [8,9,10,11,13,24,25]. The overall quality of the studies averaged eight stars (range, 7–9) on a scale from zero to nine stars (Appendix A).

The characteristics of the seven included studies and the SCFA datasets are summarized in Table 1. All of the studies were published from 1993 to 2018. Three studies were conducted in Canada [8,9,25], three in Europe [10,11,13], and one in the United States and Ghana [24]. The participants’ ages ranged from 6 to 74 years old. The overall number of obese cases was 246, and the number of nonobese controls was 198. Six studies [8,9,11,13,24,25] measured obesity using the body mass index (BMI), and one study [10] measured obesity using the BMI-Z score. Of the included studies, six measured SCFA status through the analysis of feces [8,9,10,11,13,24] and one measured SCFA status through the analysis of serum [25]. The assay method for SCFAs varied among the studies. Five studies used gas chromatography [8,9,13,24,25], one used capillary electrophoresis [10], and one used liquid chromatography [11]. Microbiology was assessed using quantitative polymerase chain reaction (qPCR) or real-time qPCR in five studies [8,9,10,13,24,25]. One study used PCR and restriction enzyme length polymorphism analysis [11]. One article reported the data stratified by level of overweight (BMI >25 kg/m^2^) and obese (BMI >30 kg/m^2^) [13]. One article reported each dataset from the United States and Ghana [24]. In addition, four of the included studies reported fecal microbiota richness in obese and nonobese individuals [8,9,10,13]. The datasets of the fecal microbiota abundance at the phylum levels are listed in Appendix A.

### 3.2. SCFAs and Obesity

According to the random-effect meta-analysis results shown in Figure 2, obese individuals had significantly higher SCFA concentrations of acetate (SMD = 0.87, 95% CI = 0.24–1.50) in the blood and feces, propionate (SMD = 0.86, 95% CI = 0.35–1.36) in feces, valerate (SMD = 0.32, 95% CI = 0.00–0.64) in feces, and butyrate (SMD = 0.78, 95% CI = 0.29–1.27) in feces than the nonobese individuals. There was no difference in the levels of total SCFAs (SMD = 0.54, 95% CI = −0.34–1.41), iso-butyrate (SMD = 0.01, 95% CI = −0.28–0.29), or iso-valerate (SMD = −0.20, 95% CI = −0.46–0.06) in the feces between the obese cases and nonobese controls. There was marked heterogeneity in total SCFA (*I*^2^ = 94.3%), acetate (*I*^2^ = 88.5), propionate (*I*^2^ = 82.3%), and butyrate (*I*^2^ = 81.7%) concentrations but not in iso-butyrate (*I*^2^ = 18.5%), valerate (*I*^2^ = 0.0%), or iso-valerate (*I*^2^ = 0.0%) concentrations.

Figure 3 shows that there was no evidence of funnel plot asymmetry. In addition, Egger’s test indicated no publication bias in total SCFAs (*p* = 0.580), acetate (*p* = 0.621), propionate (*p* = 0.580), butyrate (*p* = 0.587), iso-butyrate (*p* = 0.380), valerate (*p* = 0.495), and iso-valerate (*p* = 0.783).

We excluded four datasets with BMI-Z scores of 2.14 to 5 and one dataset with a SCFA blood sample from our subgroup analyses (Figure 4). In the 20 datasets of obese cases with a BMI >25 kg/m^2^ (Figure 4a), there was a significant increase in fecal concentrations of acetate (SMD = 1.64, 95% CI = 0.00–3.27, *I^2^* = 94.8%), propionate (SMD = 1.34, 95% CI = 0.31–2.36, *I^2^* = 88.2%), and butyrate (SMD = 1.40, 95% CI = 0.38–2.41, *I^2^* = 88.2%) in obese individuals compared to the fecal concentrations in nonobese individuals. The levels of total SCFAs (SMD = 0.58, 95% CI = −1.58–2.73, *I^2^* = 97.4%), valerate (SMD = 0.27, 95% CI = −0.39–0.43, *I^2^* = 17.6%), iso-valerate (SMD = −0.20, 95% CI = −0.62–0.22, *I^2^* = 0.0%), and iso-butyrate (SMD = 0.02, 95% CI = −0.23–0.76, *I^2^* = 41.4%) did not differ between the fecal samples of obese individuals and nonobese individuals. In the 17 datasets of obese individuals with a BMI >30 kg/m^2^ (Figure 4b), there was a significant increase in fecal concentrations of total SCFAs (SMD = 0.45, 95% CI = 0.12–0.77, *I^2^* = 0.0%), acetate (SMD = 0.34, 95% CI = 0.02–0.66, *I^2^* = 0.0%), propionate (SMD = 0.52, 95% CI = 0.20–0.85, *I^2^* = 0.0%), and butyrate (SMD = 0.34, 95% CI = 0.02–0.66, *I^2^* = 0.0%) in obese individuals compared to the fecal concentration in nonobese individuals. These subgroup analyses showed no evidence of heterogeneity among obese individuals with a BMI >30 kg/m^2^.

In the sensitivity analyses, we recalculated the combined SMD of SCFAs by omitting each study individually (Appendix A). The study-specific combined SMDs of total SCFA in feces ranged from 0.22 (95% CI = −0.52–0.95) by omission of the study by Fernandes et al, to 0.98 (95% CI = 0.32–1.65) via omission of the study by Barczyńska et al. (Appendix A). When the study by Barczyńska et al. and the study by Fernandes et al. were excluded, the SMDs of the remaining studies show a consistent positive association with no significant variation and the range of the combined SMD was narrow (Appendix A). 

### 3.3. Fecal Microbiota and Obesity

As Figure 5 shows, fecal microbiota analyses were conducted in four studies [8,9,10,13] with 21 datasets. In the stratification by phylum, we identified seven datasets for Bacteroidetes and 14 datasets for Firmicutes. Compared to the nonobese group, the abundance of Bacteroidetes (SMD = −0.36, 95% CI = −0.73–0.01) and Firmicutes (SMD = −0.10, 95% CI = −0.31–0.10) was decreased in the obese group, but the difference was not statistically significant. Statistically significant heterogeneity was found among the studies of Bacteroidetes (*I^2^* = 72.1%) and Firmicutes (*I^2^* = 58.7%). Appendix A shows no evidence of funnel plot asymmetry in Egger’s test (Bacteroidetes, *p* = 0.833; Firmicutes, *p* = 0.636).

The fecal microbiota meta-regression analyses indicated that age influenced fecal phylum microbial abundance in individuals with and without obesity (Figure 6). For older-aged participants, the Bacteroidetes concentration in feces decreased less in obese individuals than in nonobese individuals (Figure 6a). There was a significant positive association between the SMDs of Bacteroidetes concentrations in feces and participant mean ages as follows: (1)$$\mathrm{SMD}\;\mathrm{of}\;\mathrm{Bacteroidetes}\;\mathrm{concentration}\;\mathrm{in}\;\mathrm{feces}=\;-1.4218+0.0278\times \mathrm{Age}\;\left(\mathrm{years}\right)\;\mathrm{Adjusted}\;R^{2}=75.42\%,\;N=7,\;I^{2}=37.84\%,\;\tau^{2}=0.043,\;p=0.038$$

The estimated SMDs of the Bacteroidetes phylum was increased from −1.116 to −0.115 for participants aged 11 to 47 years. Furthermore, the Firmicutes phylum concentration in feces increased less in obese individuals than in nonobese individuals for older-aged participants (Figure 6b). For participants aged 38 years and older, the Firmicutes concentration in feces was decreased in obese individuals compared with that in nonobese individuals. There was a significant negative association between the SMDs of Firmicutes concentration in feces and participant mean ages as follows: (2)$$\mathrm{SMD}\;\mathrm{of}\;\mathrm{Firmicutes}\;\mathrm{concentration}\;\mathrm{in}\;\mathrm{feces}=\;1.0988-0.0291\times \mathrm{Age}\;\left(\mathrm{years}\right)\;\mathrm{Adjusted}\;R^{2}=100\%,\;N=14,\;I^{2}=0\%,\;\tau^{2}=0.043,\;p=0.038$$

The estimated SMD of phylum Firmicutes decreased from 0.779 to 0.221 for participants aged 11 to 37 years. The estimated SMD of Firumicutes decreased −0.007 to −0.269 for participants aged 38 to 47 years. The meta-regression analyses of the studies suggested that age was a possible source of heterogeneity. 

### 3.4. Quality Assessment of Included Studies

The summary of findings for the outcomes of interest and the levels of evidence using GRADE assessment are provided (Appendix A). The quality of the studies on the association between SCFA concentration and obesity was ranked as “very low” because of heterogeneity in the definition of obesity among the included studies, and imprecision of the effect estimate. In addition, the quality of the studies on the association between fecal microbiota abundance and obesity was also ranked as “very low” because of heterogeneity in fecal microbiota, participant ages, and the imprecision of the effect estimates.

## 4. Discussion

The systematic review included seven human clinical studies with a total of 246 obese cases and 198 normal controls and found differences in the levels of SCFAs in feces between the obese cases and nonobese controls. The findings show that individuals with obesity had higher fecal levels of acetate, propionate, and butyrate SCFAs, a finding that was more consistent in obese cases with a BMI >30 kg/m^2^ than in those with a BMI >25 kg/m^2^. 

The relationship between obesity and SCFAs produced by intestinal bacteria is not yet fully understood but can be explained by the following hypotheses. Intestinal anaerobic bacteria produce SCFAs, including acetate, propionate, and butyrate, as major end-products by fermenting indigestible polysaccharides [26]. These SCFAs are estimated to contribute up to approximately 200 kcal/day to the human energy balance [27] and contribute to lipogenesis and accumulation in adipocytes, leading to energy harvest [28]. In addition, higher fecal SCFA concentrations may be associated with gut dysbiosis, gut permeability, excess adiposity, and cardiometabolic risk factors [29]. Some bacterial components associated with gut dysbiosis have been implicated in the pathogenesis of obesity and various metabolic diseases by causing low-grade inflammation in adipose tissue and gut microbiota modifications. Flagellin, a structural protein of bacterial flagellum, is recognized by the Toll-like receptor (TLR) 5 [30,31]. In an animal study, a deficiency in flagellin-recognizing TLR5 was associated with obesity development and insulin resistance along with an obese-type gut microbiota [32]. The features of metabolic syndrome, including obesity, hyperlipidemia, hyperglycemia, and insulin resistance, were expressed in the recipients when an altered gut microbiota of the TLR5-deficient mice was transplanted into the intestines of wild-type germ-free mice [32]. Furthermore, the effects of SCFAs on body weight and food intake occur via G-protein coupled receptors (GPRs; GPR41, also known as free fatty acid receptor 3, GPR 43, also known as free fatty acid receptor 2) [33]. Under conditions of high carbohydrate diets and obesity, the binding of SCFAs to GPRs as signal transduction molecules might be attenuated, leading to increased intestinal energy harvesting and hepatic lipogenesis [34,35,36]. However, several lines of evidence suggest that SCFAs may be beneficial for cardiometabolic health. Lipopolysaccharide (LPS) has been associated with metabolic endotoxemia, inflammation, insulin resistance, adiposity, and hepatic fat. In an animal study, SCFAs (especially butyrate) were shown to prevent the translocation of LPS, a potent inflammatory molecule produced in the cell membrane of gram-negative bacteria [37]. SCFAs have also been shown to be involved in appetite regulation in human studies based on the finding that administrating propionate to patients with obesity led to enhanced gut hormone peptide YY and glucagon-like peptide–1 secretion with significantly reduced adiposity and overall weight gain [38]. Therefore, the role of SCFAs in obese humans requires well-designed large-scale studies in the future.

We also compared the gut bacterial richness in obese and nonobese individuals of Firmicutes and Bacteroidetes phyla, and found that obese individuals had low bacterial abundance in feces, but the differences were not statistically significant. Our finding was consistent with a previous study showing no correlation between human obesity and the proportions of Bacteroides and Firmicutes among fecal bacteria [14]. However, a recent study of the human gut microbial composition in a population sample of 123 nonobese and 169 obese individuals suggested that a decrease in the relative abundance of key bacterial species may lead to obesity [39]. Thus, the results of studies in humans have been inconsistent, generating considerable controversy as to the abundance of Bacteroides and Firmicutes and their relationship to obesity. However, our finding is valuable as it is the first meta-analysis based on human subject studies evaluating this relationship. Furthermore, we conducted a meta-regression analysis to determine whether the bacterial abundance at the phylum level was related to age. The meta-regression analysis demonstrated that the abundance of the phylum Firmicutes was positively associated with obesity for individuals with a mean age of 37 years or younger, while the abundance of the phylum Bacteroidetes was negatively associated with obesity for participants with a mean age of 47 years or younger. However, the results of this analysis should be interpreted cautiously because of the small sample size. 

This study had several strengths and limitations. Heterogeneity was found between studies when data were pooled. The potential sources of heterogeneity might be age, measurement of specimens, diet, degree of obesity, and differences in sample size. The subgroup, sensitivity, and meta-regression analyses were used to explore the potential sources of high levels of heterogeneity. The subgroup analyses showed that individuals with a BMI >30 kg/m^2^ had a higher level of total SCFAs, acetate, propionate, and butyrate without heterogeneity. The meta-regression showed that the differences in fecal microbial richness between the two groups were influenced by age. Thus, the degree of obesity and age partly account for the significant heterogeneity. We included only seven studies, some of which had a relatively small sample size. Most of the study participants were from Europe and the United States while one study was from Ghana, so our results might not be applicable to other Asian or African populations. It would be interesting to determine whether the levels of acetate, propionate, butyrate, and valerate are different between obese and nonobese individuals by sex, age, race, and diet, but the lack of data in the included studies prevented further analysis. We conducted searches based on three major electronic databases and considered only studies published in English. Some studies published in non-English languages may have been missed. Therefore, it was inevitably difficult to avoid selection and language biases in this meta-analysis. However, despite these limitations, our study has the strength of being the first meta-analysis to evaluate the relationship between SCFAs and obesity in humans, and the results of higher fecal SCFAs in obese individuals with respect to nonobese individuals are valuable. 

## 5. Conclusions

Our results indicate that obesity is associated with high levels of SCFAs but not gut microbiota richness at the phylum level. However, these results do not clearly explain the relationship due to the substantial heterogeneity and the limitations of the study designs. Additional randomized controlled studies are needed to clarify whether the role of SCFAs in energy metabolism or weight control is different between obese and nonobese individuals, but it is also necessary to identify additional contributors to SCFA production, absorption, and excretion in humans.

## Figures and Tables

**Figure 1 nutrients-11-02512-f001:**
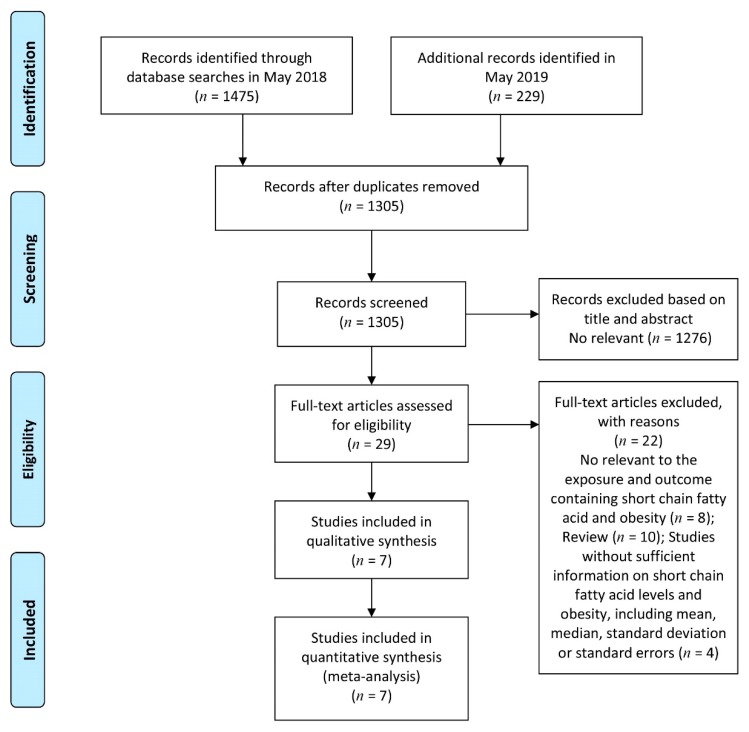
Flow diagram of the search strategy and study selection process.

**Figure 2 nutrients-11-02512-f002:**
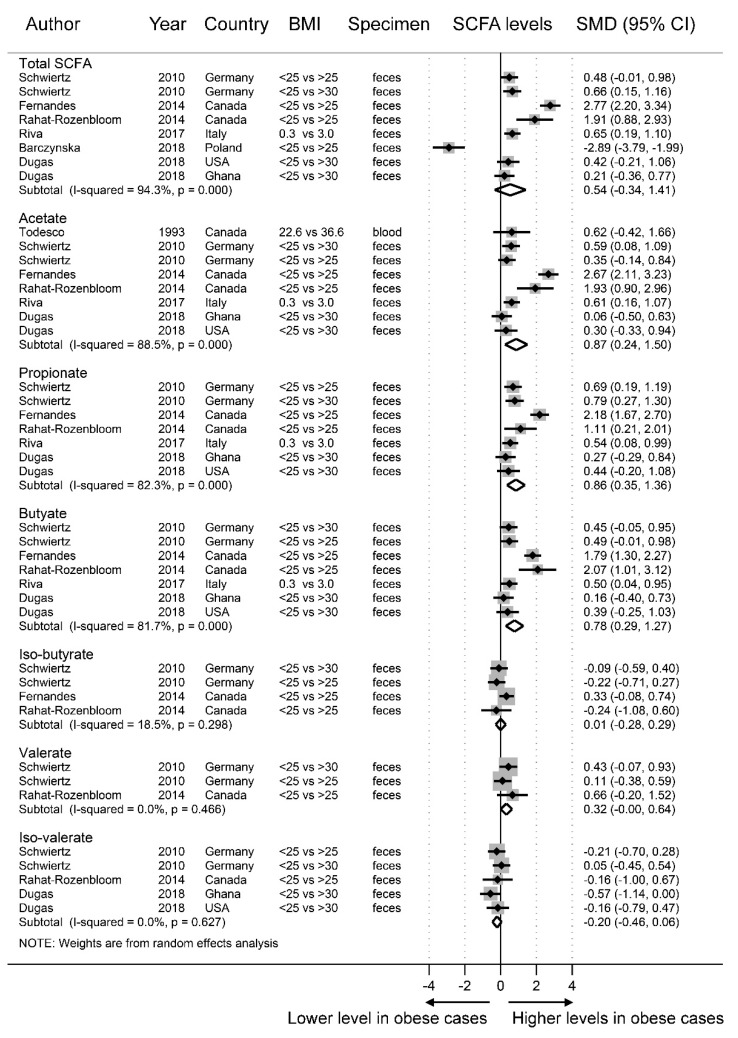
Forest plots of studies of short-chain fatty acid (SCFA) levels in obese and non-obese individuals. The combined standardized mean differences (SMD) and 95% confidence intervals (CIs) were calculated using random-effects models.

**Figure 3 nutrients-11-02512-f003:**
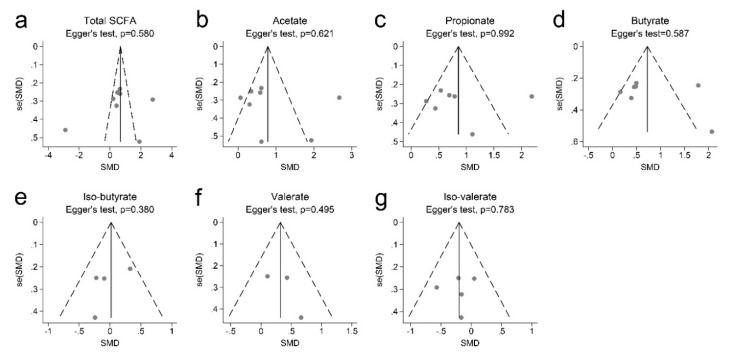
Begg’s funnel plots with 95% confidence intervals for the meta-analysis of SCFAs and obesity. (**a**) total SCFA; (**b)** acetate; (**c**) propionate; (**d**) butyrate; (**e**) iso-butyrate; (**f**) valerate; (**g**) iso-valerate.

**Figure 4 nutrients-11-02512-f004:**
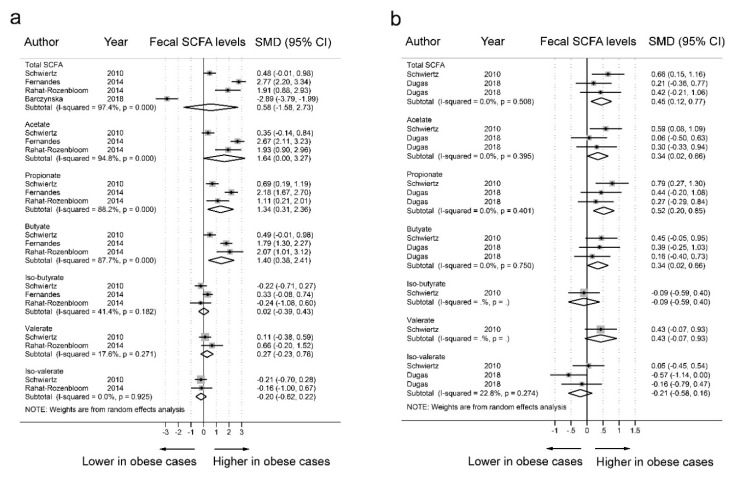
Subgroup analyses of the fecal short-chain fatty acid (SCFA) levels in obese and nonobese individuals. **a**. Forest plots of 20 data sets of SCFA levels in obese cases (BMI >25kg/m^2^) and nonobese controls (BMI <25kg/m^2^); **b**. Forest plots of 17 datasets of SCFA levels in obese cases (BMI >30kg/m^2^) and nonobese controls (BMI <25kg/m^2^). The combined standardized mean differences (SMDs) and 95% confidence intervals (CIs) were calculated using random effects models.

**Figure 5 nutrients-11-02512-f005:**
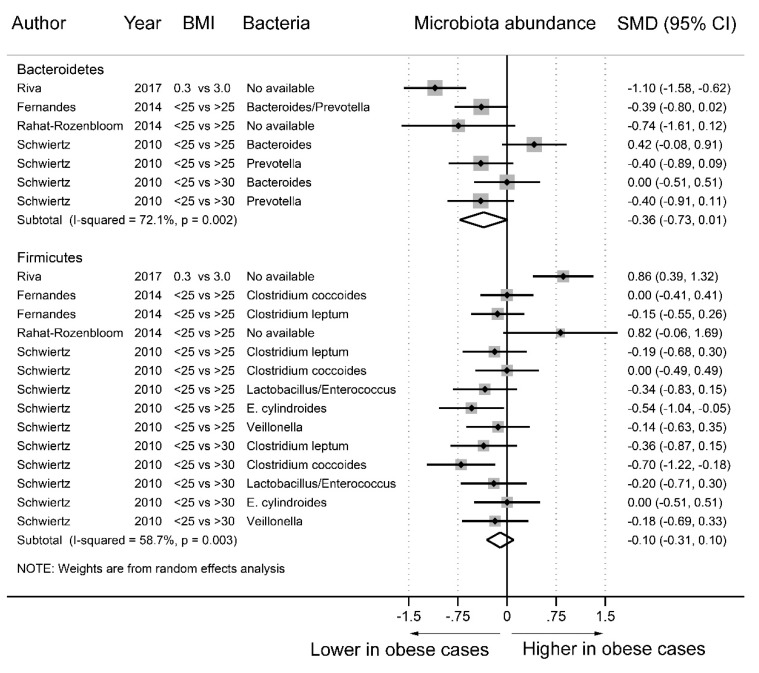
Forest plots of studies of fecal microbiota abundance at the phylum level in obese and nonobese individuals. The combined standardized mean differences (SMD) and 95% confidence intervals (CIs) were calculated using random effects models.

**Figure 6 nutrients-11-02512-f006:**
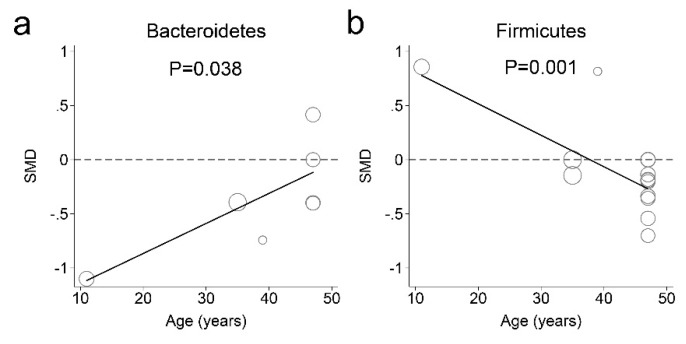
Association between age and the fecal microbiota abundance in obese and nonobese individuals. The participants’ mean ages were modeled using separate random-effects meta-regression models. The fecal microbiota ((**a**) Bacteroidetes; (**b**) Firmicutes) abundance levels were compared between obese and nonobese individuals. The Y axes indicate a standardized mean difference (SMD). Each circle represents a study and the size of the circle reflects the influence of that study on the model. The circle size is inversely proportional to the standard error of that study. The solid lines represent the weighted regression lines based on variance-weighted least squares.

**Table 1 nutrients-11-02512-t001:** Characteristics of studies included in the analysis of short chain fatty acid (SCFA) levels.

BMI Category,	Sex, Age	SCFA	Specimen	Obese Individuals			Nonobese Individuals			Measure
Study	(mean, y)		unit	*n*	Mean	SD	*n*	Mean	SD	SCFA
BMI-Z (mean)				2.14–5 (3)			−2.12–1.56 (0.3)			
Riva,	M and F	Total SCFA	Feces	42	65.3	32.4	36	47.5	20.4	CE
2017, Italy	9–16 (11)	Acetate	µmol/g	42	40.4	18.9	36	30.3	13	
		Propionate		42	12.5	7.7	36	8.8	5.8	
		Butyrate		42	12.4	9.8	36	8.4	5.3	
BMI (kg/m^2^)				>25			<25			
Barczyńska,	M and F	Total SCFA	Feces	20	3.59	0.49	20	5.44	0.76	HPLC
2018, Poland	6–15 (10)		mg/g							
BMI (kg/m^2^)				>25			<25			
Fernandes,	M and F	Total SCFA	Feces	42	89.7	4.2	52	77.6	4.5	GC
2014, Canada	18–67 (35)	Acetate	mmol/kg	42	48	2.3	52	41.4	2.6	
		Propionate		42	17.6	1.2	52	15.1	1.1	
		Butyrate		42	16.1	1	52	14	1.3	
		Iso-butyrate		42	3	0.4	52	2.9	0.2	
BMI (kg/m^2^)				>25			<25			
Rahat-Rozenbloom,	M and F	Total SCFA	Feces	11	81.3	7.4	11	64.1	10.4	GC
2014, Canada	17 < (39)	Acetate	mmol/kg	11	45.3	4.3	11	35.1	6.1	
		Propionate		11	15.4	2	11	12.7	2.8	
		Butyrate		11	15.4	1.7	11	11.1	2.4	
		Iso-butyrate		11	1.4	0.3	11	1.5	0.5	
		Valerate		11	1.9	0.4	11	1.6	0.5	
		Iso-valerate		11	2	0.5	11	2.1	0.7	
BMI (kg/m^2^)				>25			<25			
Schwiertz,	M and F	Total SCFA	Feces	35	98.7	33.9	30	84.6	22.9	GC
2010, Germany	14–74 (47)	Acetate	mmol/L	35	56	18.2	30	50.5	12.6	
		Propionate		35	18.3	7.9	30	13.6	5.2	
		Butyrate		35	18.5	10.1	30	14.1	7.6	
		Iso-butyrate		35	1.6	0.9	30	1.8	0.9	
		Valerate		35	2	1.1	30	1.9	0.7	
		Iso-valerate		35	2.3	1.7	30	2.7	2.1	
BMI (kg/m^2^)				>30			<25			
		Total SCFA		33	103.9	34.3	30	84.6	22.9	
		Acetate		33	59.8	18.3	30	50.5	12.6	
		Propionate		33	19.3	8.7	30	13.6	5.2	
		Butyrate		33	18.1	10	30	14.1	7.6	
		Iso-butyrate		33	1.7	1.2	30	1.8	0.9	
		Valerate		33	2.3	1.1	30	1.9	0.7	
		Iso-valerate		33	2.8	2	30	2.7	2.1	
BMI (kg/m^2^)				>30			<25			
Dugas,	F	Total SCFA	Feces	21	5.48	1.35	29	5.09	2.19	GC/MC
2018, Ghana	25–45	Acetate	µg/mL	21	2.12	0.44	29	2.08	0.73	
		Propionate		21	1.28	0.49	29	1.11	0.7	
		Butyrate		21	1.79	0.8	29	1.65	0.89	
		Iso-butyrate		21	0.09	0.04	29	0.12	0.06	
BMI (kg/m^2^)				>30			<25			
Dugas,		Total SCFA		37	3.76	1.94	13	3.01	1.13	
2018, USA		Acetate		37	1.6	0.74	13	1.39	0.53	
		Propionate		37	0.71	0.43	13	0.54	0.22	
		Butyrate		37	1.18	0.85	13	0.85	0.57	
		Iso-butyrate		37	0.1	0.07	13	0.11	0.03	
BMI (mean, kg/m^2^)				36.6			22.6			
Todescol,	M and F	Acetate	Blood	8	3.6	1.4	7	2.9	0.7	GC
1993, Canada	O (33.4),		µmol/dL							
	NO (29.0)									

M, male; F, Female; O, y, year; obese; NO, nonobese; BMI, body mass index; GC, gas chromatography; MS, mass spectrometry; HPLC, high performance liquid chromatography; LC, liquid chromatography; OTU, operational taxonomic unit; RT-qPCR, real time quantitative polymerase chain reaction; RFLP, restriction enzyme length polymorphism analysis; SCFA, short chain fatty acid; SD, standard deviation.

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
