# Peer review of "Short Chain Fatty Acids and Fecal Microbiota Abundance in Humans with Obesity: A Systematic Review and Meta-Analysis"

_nutrients, 2019, doi:10.3390/nu11102512_

Round 1
Reviewer 1 Report
The manuscript is focusing on an interesting aspect of the more and more emerging studies on obesity and the related diseases and potential health risks. The authors have not performed a yet another study but screened the existing literature and tried to systematize the outcome. This approach is relevant and could bring in some needed agreement within the literature sources. However, there is further room for improvement into the manuscript.
The selection of the relevant studies is described and one can easily follow the data selection, however, it is absolutely not commented on how different data is compared. For example, there are different analytical techniques used within the different selected studies which measure the amounts of the short chain fatty acids at different phases (gas chromatography vs HPLC, for example is one of the extremes. The measurements are also performed on blood or at the faeces, whose concentrations in Table 1 are given in different measurement units (micromol/g, mg/g, mmol/kg, mmol/l, microg/ml) and therefore the reader cannot clearly understand the content. Please, correct the values such that they are uniform.
This discrepancy brings in as well the question which values were then used in the following statistical evaluations and this question the outcome. Could you, please, explain this point?
Additionally to the design of Table 1: the abbreviation N is used twice: once to indicate N as nonobese (as abbreviated below the table) and once to indicate what looks like a number N in the columns. CE - as capillary electrophoresis is not abbreviated below the table.
If we assume that the discrepancy in the provided in the Table 1 values has been taken into account and therefore correct statistical evaluation of the studies has been performed, there are some problems with the provided results in the Figure 2 with the Forest plots. There is a clear outlier for the total SCFA with the study from Barczynska, which was not used in the following separate plots for the SCFA. There is the need to provide the respective Forest plot when this data is excluded as it is said that the data is excluded in the Results section.
Similarly, the studies of Fernandez and Rahat-Rozenbloom show variations as compared with the other studies. The authors have not commented on that and have not provided any potential reason either originating from the dietary, age or other reasons, or if this could be attributed to geographical origin. This comes relevant to the fact that they comment of having a single study from Ghana and thus bringing in the geographically related differences. However, the Ghana outcome is much closer to the outer studies than the ones from Canada. Could you, please, refer to this.
Author Response
Reviewer #1
1. The selection of the relevant studies is described and one can easily follow the data selection, however, it is absolutely not commented on how different data is compared. For example, there are different analytical techniques used within the different selected studies which measure the amounts of the short chain fatty acids at different phases (gas chromatography vs HPLC, for example is one of the extremes. The measurements are also performed on blood or at the faeces, whose concentrations in Table 1 are given in different measurement units (micromol/g, mg/g, mmol/kg, mmol/l, microg/ml) and therefore the reader cannot clearly understand the content. Please, correct the values such that they are uniform. This discrepancy brings in as well the question which values were then used in the following statistical evaluations and this question the outcome. Could you, please, explain this point?
Response to reviewer #1. 1:
We used only published data. The use of this method led to limited availability of data than individual patient data meta-analysis. Therefore, we used the standardized mean difference (SMD).
Please refer to the following paragraphs for a description of SMD in the Cochran booklet (https://handbook-5-1.cochrane.org/chapter_9/9_2_3_2_the_standardized_mean_difference.htm).
“The SMD is used as a summary statistic in meta-analysis when the studies all assess the same outcome but measure it in a variety of ways (for example, all studies measure depression but they use different psychometric scales). In this circumstance it is necessary to standardize the results of the studies to a uniform scale before they can be combined. The standardized mean difference expresses the size of the intervention effect in each study relative to the variability observed in that study. (Again in reality the intervention effect is a difference in means and not a mean of differences.):
Thus studies for which the difference in means is the same proportion of the standard deviation will have the same SMD, regardless of the actual scales used to make the measurements.”
2. Additionally to the design of Table 1: the abbreviation N is used twice: once to indicate N as nonobese (as abbreviated below the table) and once to indicate what looks like a number N in the columns. CE - as capillary electrophoresis is not abbreviated below the table.
Response to reviewer #1. 2:
We have modified the abbreviation used as follows: NO, Nonobese
3.If we assume that the discrepancy in the provided in the Table 1 values has been taken into account and therefore correct statistical evaluation of the studies has been performed, there are some problems with the provided results in the Figure 2 with the Forest plots. There is a clear outlier for the total SCFA with the study from Barczynska, which was not used in the following separate plots for the SCFA. There is the need to provide the respective Forest plot when this data is excluded as it is said that the data is excluded in the Results section.
Response to reviewer #1. 3:
We used only published data. The data of interest, presented as continuous values (means and SDs), were used to perform the meta-analysis to obtain the standardized mean differences (SMDs) and 95% confidence intervals (CIs) of the SCFA levels of participants with obesity and controls.
The study of Barczynska et al. provided the estimates (N, mean, SD) of total SCFA but not provided the SDs of separate SCFA. So we could use only the data of total SCFA for data synthesis and statistical analysis.
4. Similarly, the studies of Fernandez and Rahat-Rozenbloom show variations as compared with the other studies. The authors have not commented on that and have not provided any potential reason either originating from the dietary, age or other reasons, or if this could be attributed to geographical origin. This comes relevant to the fact that they comment of having a single study from Ghana and thus bringing in the geographically related differences. However, the Ghana outcome is much closer to the outer studies than the ones from Canada. Could you, please, refer to this.
Response to reviewer #1. 4:
There were no data on subject diets in the included studies. Future studies on diet, obesity, and SCFA may be necessary to examine contributing factors to association between SCFA and obesity. In addition, in the discussion of this study, we explained as much as possible the reasons for heterogeneity among the studies as possible as follows:
“This study had several strengths and limitations. Heterogeneity was found between studies when data were pooled. The potential sources of heterogeneity might be age, measurement of specimens, diet, degree of obesity and differences in sample size. The subgroup, sensitivity and meta-regression analyses were used to explore the potential sources of high levels of heterogeneity. The subgroup analyses showed that individuals with a BMI >30 kg/m2 had a higher level of total SCFAs, acetate, propionate, and butyrate without heterogeneity. The meta-regression showed that the differences in fecal microbial richness between the two groups were influenced by age. Thus, the degree of obesity and age partly account for the significant heterogeneity. We included only 7 studies, some of which had a relatively small sample size. Most of the study participants were from Europe and the United States, while one study was from Ghana, so our results might not be applicable to other Asian or African populations. It would be interesting to determine whether the levels of acetate, propionate, butyrate and valerate are different between obese and nonobese individuals by sex, age, race and diet, but the lack of data in the included studies prevented further analysis. We conducted searches based on three major electronic databases and considered only studies published in English. Some studies published in non-English languages may have been missed. Therefore, it was inevitably difficult to avoid selection and language biases in this meta-analysis. However, despite these limitations, our study has the strength of being the first meta-analysis to evaluate the relationship between fecal SCFAs and obesity in humans, and the results of higher fecal SCFAs in obese individuals with respect to nonobese individuals are valuable.”
Reviewer 2 Report
Dear Authors
the topic of the paper is relevant and original.
The paper is well presented and organized
Author Response
Thank you for reviewing the manuscript.
Reviewer 3 Report
The authors described well regarding the limitations of the current study in the Discussion section. Therefore, despite the small sample sizes and biased regional selections (Europe vs. Asia), the study displays its merit(s).
The primary concern of this manuscript is that lack of full description of the criteria the authors used, including ‘Inclusion and exclusion criteria’.
In addition, although the authors performed the ‘Literature search’ based on the two references (references #16 and #17), which are dated 2008 and 2009, the overall criteria of the literature search were not well-described. Thus, it is necessary to describe how/what was excluded in the n=1276 criteria (to end up n=29).
Due to this, despite the sound analyses performed by the authors, the sample size has become very small to make any statistically significant conclusion. The authors did point out the limitations of the study at the Discussion section (mainly Europe), however, more detailed and clear description of the criteria is necessary.
Author Response
Reviewer #3.
The authors described well regarding the limitations of the current study in the Discussion section. Therefore, despite the small sample sizes and biased regional selections (Europe vs. Asia), the study displays its merit(s).
1.The primary concern of this manuscript is that lack of full description of the criteria the authors used, including ‘Inclusion and exclusion criteria’.
Response to reviewer #3. 1
We planned, conducted, and reported this systematic review according to widely accepted quality standards for reporting Meta-analyses of Observational Studies in Epidemiology and Preferred Reporting Items for Systematic Reviews and Meta-Analyses guidelines [16,17].
In the section of inclusion and exclusion, the full information has been included on the study design, patient characteristics, publication status, and language used as follows.
“Inclusion and exclusion criteria
Published articles were included in this meta-analysis if they met the following criteria: (1) case-control studies that were conducted in humans rather than animals; (2) studies that provided data on SCFA levels in individuals both with and without obesity; and (3) studies that were written in English and published in their entirety. The exclusion criteria in this meta-analysis were the following: (1) articles that did not satisfy the inclusion criteria; (2) publication types such as animal studies, reviews, case reports, and systematic reviews; and (3) studies that did not provide sufficient data on SCFA levels, including means, medians, standard deviations (SDs), and/or standard errors for individuals with and without obesity.”
2. In addition, although the authors performed the ‘Literature search’ based on the two references (references #16 and #17), which are dated 2008 and 2009, the overall criteria of the literature search were not well-described.Thus, it is necessary to describe how/what was excluded in the n=1276 criteria (to end up n=29).
Response to reviewer #3. 2
Among the studies retrieved in the search, we removed duplicate studies, selected studies that met the inclusion/exclusion criteria based on the abstract and title, and then made the final selection of studies based on their full text.
We added the description in the first ground of selection section step based on the abstract and title and as follows: " Records excluded based on the title and abstract: No relevant (n=1276)"
3. Due to this, despite the sound analyses performed by the authors, the sample size has become very small to make any statistically significant conclusion.
Response to reviewer #3. 3
The systematic review included 7 human clinical studies with a total of 246 obese cases and 198 normal controls and found differences in the levels of SCFAs in feces between the obese cases and nonobese controls. Thus, considering the characteristics of the meta-analysis study, which analyzes the data reported in the published study, it is considered that the number of subjects and the number of study are adequate to analyze the relationship between obesity and short-chain fatty acids.
4. The authors did point out the limitations of the study at the Discussion section (mainly Europe), however, more detailed and clear description of the criteria is necessary.
Response to reviewer #3. 4
We described the limitation of the study regarding the size of sample and studies included in this meta-analysis at the discussion section as follows.
“We included only 7 studies, some of which had a relatively small sample size. Most of the study participants were from Europe and the United States, while one study was from Ghana, so our results might not be applicable to other Asian or African populations. It would be interesting to determine whether the levels of acetate, propionate, butyrate and valerate are different between obese and nonobese individuals by sex, age, race and diet, but the lack of data in the included studies prevented further analysis. We conducted searches based on three major electronic databases and considered only studies published in English. Some studies published in non-English languages may have been missed. Therefore, it was inevitably difficult to avoid selection and language biases in this meta-analysis. However, despite these limitations, our study has the strength of being the first meta-analysis to evaluate the relationship between fecal SCFAs and obesity in humans, and the results of higher fecal SCFAs in obese individuals with respect to nonobese individuals are valuable.”
Round 2
Reviewer 1 Report
Together with the answers to the reviewers the manuscript has improved. However, the explanation provided by the authors in their response about the standard mean differences (SMD) must be included in the manuscript as this makes it clear to the reader. Without this explanation questions arise which are difficult for not so experienced in statistics reader cannot easily answer.
Author Response
Comments and Suggestions for Authors and Response to the comments
Reviewer #1
Together with the answers to the reviewers the manuscript has improved. However, the explanation provided by the authors in their response about the standard mean differences (SMD) must be included in the manuscript as this makes it clear to the reader. Without this explanation questions arise which are difficult for not so experienced in statistics reader cannot easily answer.
Response to reviewer #1
We have modified the description as follows:
“The data of interest, presented as continuous values (means and SDs), were used to perform the meta-analysis to obtain the standardized mean differences (SMDs) and 95% confidence intervals (CIs) of the SCFA levels of participants with obesity and controls. The SMDs were calculated by subtracting the means of SCFAs levels between the two groups and dividing by the standard deviations. Thus, studies for which the difference in means is the same proportion of the standard deviation will have the same SMD, regardless of the actual scales used to make the SCFAs measurements.”